# *Let's Roll a BiFTA*: Bi-refinement for Fine-grained Text-visual Alignment in Vision-Language Models

## Abstract

Recent research has shown that aligning fine-grained text descriptions with localized image patches can significantly improve the zero-shot performance of pre-trained vision-language models (e.g., CLIP). However, we find that both fine-grained text descriptions and localized image patches often contain redundant information, making text-visual alignment less effective. In this paper, we tackle this issue from two perspectives: *view refinement* and *description refinement*, termed as $\boldsymbol{Bi}$-*refinement for* $\boldsymbol{F}$*ine-grained* $\boldsymbol{T}$*ext-visual* $\boldsymbol{A}$*lignment* (BiFTA). *View refinement* removes redundant image patches with high *Intersection over Union* (IoU) ratios, resulting in more distinctive visual samples. *Description refinement* removes redundant text descriptions with high pairwise cosine similarity, ensuring greater diversity in the remaining descriptions. BiFTA achieves superior zero-shot performance on 6 benchmark datasets for both ViT-based and ResNet-based CLIP, justifying the necessity to remove redundant information in visual-text alignment. Our code is available at: `https://anonymous.4open.science/r/BiFTA-A707`.

## 1 Introduction

Drawing from the profound strides made in large-scale pre-training within natural language processing (Radford et al., 2018; 2019; Devlin et al., 2019; Brown et al., 2020), the CLIP model (Radford et al., 2021) aligns vast collections of images with their corresponding natural language captions (e.g., "a photo of a {label}") into a unified embedding space using large datasets. The scaling of the pre-training data in CLIP empowers the model to deliver considerable performance in zero-shot classification (Radford et al., 2021). To push the limits of CLIP's zero-shot capabilities, several studies (Menon & Vondrick, 2023; Pratt et al., 2023) harness the power of LLMs to craft detailed, fine-grained textual descriptions for each label category by using label-integrated prompt templates (e.g., "describe what does a/an {label} look like"), achieving a more precise alignment between an entire image and descriptive textual representations. Building upon this, Li et al. (2024) propose $\boldsymbol{w}$*eighted visual-text* $\boldsymbol{c}$*ross* $\boldsymbol{a}$*lignment* (WCA), which aligns these fine-grained textual descriptions with localized image patches based on weighted similarities to further refine the synergy between visual and textual representations, achieving *state-of-the-art* (SOTA) zero-shot performance.

However, we find that both fine-grained textual descriptions and localized image patches often contain redundant information, as demonstrated in Figure 1, making text-visual alignment less effective. For example, following WCA (Li et al., 2024), we use random cropping to obtain localized image patches of an image sample (e.g., a border collie), as depicted in Figure 1. We find that these image patches often include certain redundant views exhibiting exceptionally high pairwise cosine similarities (i.e., approaching to 1), which is attributed to the randomness in the cropping method. Similarly, we observe that the diversity of LLM-generated textual descriptions is often restricted by the invariant label-integrated prompt template, leading to redundant descriptions (see Figure 1). These redundant views and descriptions can cause significant biases when computing the similarity scores, since they are *overrepresented*. Consequently, the CLIP model may overemphasize duplicated features, and thus skew the result toward the redundant features. These observations motivate us to remove redundant information within these image patches and textual descriptions.

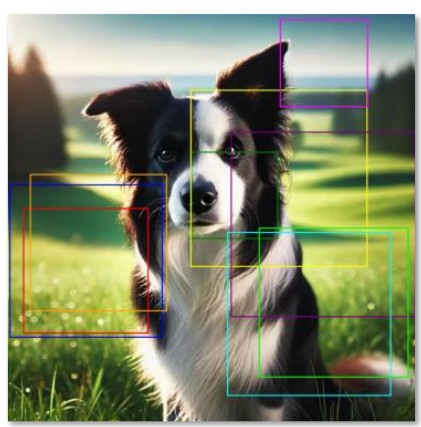 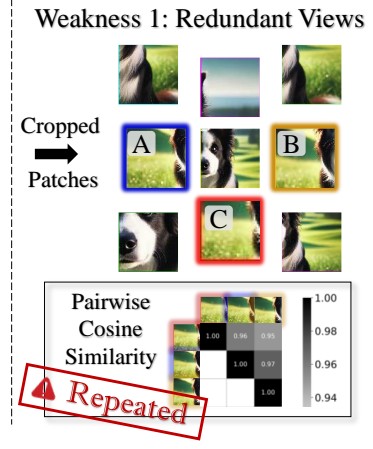 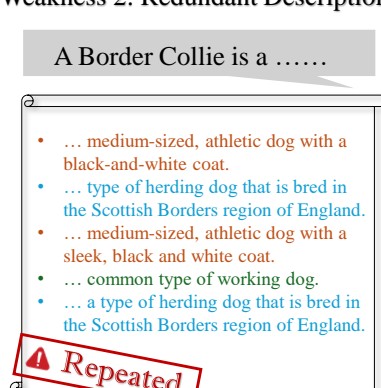

Figure 1: Weaknesses of weighted visual-text cross alignment (Li et al., 2024). **Weakness 1: Pairwise similarity scores of highly overlapping crop bounding boxes.** We demonstrate that image patches A, B, and C, exhibiting significant overlap and redundancy, which provide limited semantic information and consequently contribute minimally to accurate classification. **Weakness 2: Redundant textual descriptions generated by LLM.** We gather textual descriptions from previous work and demonstrate that a significant portion of these descriptions are redundant for a given category, thereby diluting the contribution of meaningful and informative descriptions.

To this end, we propose ***Bi***-*refinement for* ***F***ine-grained ***T***ext-visual ***A***lignment (BiFTA), a new method to tackle the above-mentioned issue from two perspectives: *view refinement* and *description refinement*. Specifically, *view refinement* uses *Intersection over Union* (IoU) as the filter metric to efficiently identify and eliminate redundant cropped image patches based on their overlaps of the bounding box. We aim to remove image patches with high IoU ratios, making the remaining visual samples more distinctive. In contrast, *description refinement* first computes the pairwise cosine similarity of the textual descriptions at the representation level, aiming to filter out redundant ones. Then, we select top-$k$ textual descriptions that have the highest cosine similarities with the label caption ("a photo of a/an {label}") from the remaining ones. To better understand BiFTA, we provide theoretical justifications in Section 4.3 to show that our method indeed prioritizes more effective image patches and descriptions while filtering out redundant information.

Through extensive evaluations across 6 benchmark datasets, we show that BiFTA outperforms baseline methods by notably improving the zero-shot classification accuracy for both ViT-based and ResNet-based CLIP, justifying the necessity to remove redundant information in visual-text alignment.

We summarize the main contributions of our work as follows:

- We observe that localized image patches and fine-grained textual descriptions often contain redundant information, making current SOTA visual-text alignment methods less effective.

- We propose a new method, namely BiFTA, to mitigate such redundancy through *view refinement* and *description refinement*, enhancing the alignment between visual and textual modalities.

- We empirically show that BiFTA outperforms baseline methods by achieving significant improvements in zero-shot classification accuracy across 6 benchmark datasets with various CLIP backbones.

## 2 Related Work

**Zero-shot learning for Vision-Language Models.** *Vision-language models* (VLMs) have shown their emergent capabilities on image captioning, visual question answering and image classification, which are

not specifically pre-trained or explicitly finetuned for these downstream tasks (Radford et al., 2021; Cho et al., 2021; Wang et al., 2021; Kim et al., 2021; Xue et al., 2021; Li et al., 2022a; Alayrac et al., 2022). CLIP (Radford et al., 2021) demonstrates that integrating large-scale pre-training on image-text pairs with a contrastive loss function could enable zero-shot transfer to downstream tasks by simply using natural language prompts. Similarly, ALIGN (Jia et al., 2021) further demonstrates robust representation learning capability of VLMs by pre-training on noisy image-text pairs at large scale. By increasing the scale of the pre-training data and model size, Florence (Yuan et al., 2021) introduces a unified vision-language foundation model capable of zero-shot image classification and retrieval. On the other hand, CoCa (Yu et al., 2022) combines contrastive and generative objectives to improve zero-shot generalization across diverse tasks. The scaling of pre-training data and the contrastive learning paradigm contribute to deeper visual-text alignment and visual understanding of the model.

**Textual prompt engineering in VLMs.** By scaling the training data, VLMs can learn and understand diverse visual concepts, which can then be transferred to downstream tasks through specific textual label prompting (Radford et al., 2021; Jia et al., 2021; Yuan et al., 2021; Li & Liang, 2021; Singh et al., 2022; Zhou et al., 2022; Shu et al., 2022; Cui et al., 2025). The LLM-integrated textual description generation shows a great generalization ability comparing with existing prompt-learning methods, which often overfit to training data (Li et al., 2022b; Wang et al., 2022; Wu et al., 2023; Tanwisuth et al., 2023). CLIP (Radford et al., 2021) achieves zero-shot classification by generating classification weights through encoding textual descriptions that uses CLIP template and categories via its text encoder. It then compares these text embeddings with image features extracted by the image encoder to determine the most likely class. Zhou et al. (2022) discover that manually prompt tuning is a time-consuming task and propose CoOp, which models context words with continuous vectors. Subsequently, Menon & Vondrick (2023); Pratt et al. (2023) automatically generates textual category-specific descriptions by leveraging LLMs with different prompt templates. These textual descriptions can accurately reflect visual features of images in each category. More recently, *retrieval-augmented generation* (RAG) is proposed to help to generate accurate descriptions of categories, which is a training-free framework that can be directly integrated during inference time (Yu et al., 2024; Chan et al., 2024; Guo et al., 2024). It retrieves semantically relevant documents by computing embedding vector similarity and provides the retrieved information as supplementary context to LLMs, enabling more accurate and informed responses.

**Fine-grained visual-text alignment.** *Weighted visual-text cross alignment* (WCA) (Li et al., 2024) has done empirical observation on the embedding alignment between visual patches and textual descriptions. It uses random crops to augment image samples and utilizes cosine similarity to extract informative patches. Similarly, it utilizes distinctive textual descriptions from LLM to cross-align with the image patches. There are uncertainties when cropping samples randomly, and the textual descriptions are not corresponding to fine-grained image patches. AttrVR (Cai et al., 2025) uses descriptive and distinctive attributes of each categories from LLM outputs. In our method, we augment the textual descriptions by integrating various description generation methods to further select the high-quality textual descriptions.

## 3 Preliminary

**CLIP Zero-shot Classification.** CLIP (Radford et al., 2021) is a pre-trained VLM that consists of a text encoder $f_{\text{txt}} : \mathcal{T} \to \mathcal{Z}$ and an image encoder $f_{\text{img}} : \mathcal{X} \to \mathcal{Z}$, where $\mathcal{T}$ is a discrete text space, $\mathcal{X}$ is a continuous image space and $\mathcal{Z} \subseteq \mathbb{R}^n$ is a shared $n$-dimensional embedding space. These encoders take an image $X \in \mathcal{X}$ and a text $T \in \mathcal{T}$ as input pair $(X, T)$, mapping them into the shared latent space $\mathcal{Z}$. Then the similarity score between the image and text embedding is calculated as:

$$\text{sim}_{\text{CLIP}}(X, T) = \cos\left(Z_i, Z_t\right) / \tau, \text{ with } Z_i = f_{\text{img}}(X), \text{ and } Z_t = f_{\text{txt}}(T),$$

where $\cos(\cdot, \cdot)$ denotes the cosine similarity such that $\cos(Z_i, Z_t) = \frac{Z_i \cdot Z_t}{\|Z_i\| \|Z_t\|}$ and $\tau$ is a temperature parameter.

For downstream classification tasks, the text encoder of CLIP model receives a label prompt string $\hat{T}_y$ (e.g., "This is a photo of a/an $[y]$"), where $y \in \mathcal{Y}$. Subsequently, CLIP model predicts the label which maximizes

the probability of $p_{\text{CLIP}}(Y|X)$, given by:

$$\arg\max_{y \in \mathcal{Y}} p_{\text{CLIP}}(Y = y \mid X) = \frac{\exp\left(\text{sim}_{\text{CLIP}}(X, \hat{T}_y)\right)}{\sum_{y' \in \mathcal{Y}} \exp\left(\text{sim}_{\text{CLIP}}(X, \hat{T}_{y'})\right)}.$$

Here, the label $y$ that maximizes the conditional probability $p_{\text{CLIP}}(Y = y \mid X)$ will be chosen, where the visual and textual representations exhibit the highest similarity within the embedding space $\mathcal{Z}$. This enables zero-shot classification capability on CLIP model, as it can generalize to unseen categories without additional fine-tuning.

**Weighted Cross-Alignment (WCA).** WCA is a scoring method specifically designed to improve the visual-text cross-alignment capability of the CLIP model (Li et al., 2024). First, an original image $X = x$ is randomly cropped with a window size ranging from $[\alpha, \beta] \in [0, 1]$ for $n$ times. It obtain $n$ cropped image patches denote as $I_i = \text{rnd\_crop}(x), i \in [0, n]$, where $\text{rnd\_crop}(\cdot)$ is the random cropping function that obtains localized visual features (Li et al., 2024). On the other hand, WCA would prepare $m$ textual descriptions $T_1, T_2, ..., T_m$ generated by LLMs, which encompass descriptive features of each category $y \in \mathcal{Y}$. The textual descriptions are collected by leveraging a LLM with manually crafted prompt templates, such as "Describe what a/an category looks like." (Pratt et al., 2023). Then the visual-text similarity score matrix can be presented as:

$$\begin{bmatrix} \text{sim}_{\text{CLIP}}(I_1, T_1) & \cdots & \text{sim}_{\text{CLIP}}(I_1, T_m) \\ \vdots & \ddots & \vdots \\ \text{sim}_{\text{CLIP}}(I_n, T_1) & \cdots & \text{sim}_{\text{CLIP}}(I_n, T_m) \end{bmatrix},$$

where $\text{sim}_{\text{CLIP}}(I_i, T_j)$ represents the similarity scores between a specific textual description $T_j$ and an image patches $I_i$. When applying WCA, the overall similarity score between image $x$ and label $y$ is as follows:

$$\text{sim}_{\text{WCA}}(X = x, Y = y) = \sum_{i=1}^{n} \sum_{j=1}^{m} w_i v_j \text{sim}_{\text{CLIP}}(I_i, T_j), \tag{1}$$

where $w_i$ and $v_j$ are weights for image patch $I_i$ and textual description $T_j$, respectively. They are obtained from the similarity between $I_i$ and the original image $x$, or $T_j$ and the label prompt string $\hat{T}_y$, i.e., $w_i = \text{softmax}(\cos(f_{\text{img}}(x), f_{\text{img}}(I_i)))$ and $v_j = \text{softmax}(\cos(f_{\text{txt}}(\hat{T}_y), f_{\text{txt}}(T_j)))$. WCA effectively aligns visual-text pairs with aforementioned weights (Li et al., 2024). However, redundant information may appear in image patches and textual descriptions, leading to the weaknesses shown in Figure 1.

## 4 BiFTA: View Refinement and Description Refinement

We introduce BiFTA from two key perspectives: (1) *view refinement* (Section 4.1), which involves filtering out localized image patches generated through random cropping; and (2) *description refinement* (Section 4.2), which encompasses strategies designed to collect fine-grained textual descriptions. An overview of BiFTA is illustrated in Figure 2. Our approach integrates efficient data filtering and refinement techniques to refine cross-aligned image-text pairs, ultimately enhancing the quality and accuracy of image classification. Furthermore, in Section 4.3, we mathematically prove that BiFTA generates more effective views and descriptions by leveraging posterior probability analysis.

### 4.1 View Refinement

As shown in Section 3, in WCA, for a single image $x$, a set of randomly selected patches $V = \{I_1, I_2, ..., I_n\}$ is used for subsequent classification. However, as illustrated in Figure 1, this may introduce redundancy, affecting the classification results. Therefore, without changing the size of the patch queue $|V|$, we employ a filtering function $f_{\text{IoU}}(\cdot)$ to ensure that each newly cropped image added to current $V$ does not have excessive overlap with the existing images in the set. For a newly cropped image $I_i = \text{rnd\_crop}(x)$, the filtering function can be expressed as:

$$f_{\text{IoU}}(I_i, V) = \begin{cases} 1 & \forall I \in V, \text{IoU}(I, I_i) < 1 - \delta \\ 0 & \exists I \in V, \text{IoU}(I, I_i) \geq 1 - \delta \end{cases}, \tag{2}$$

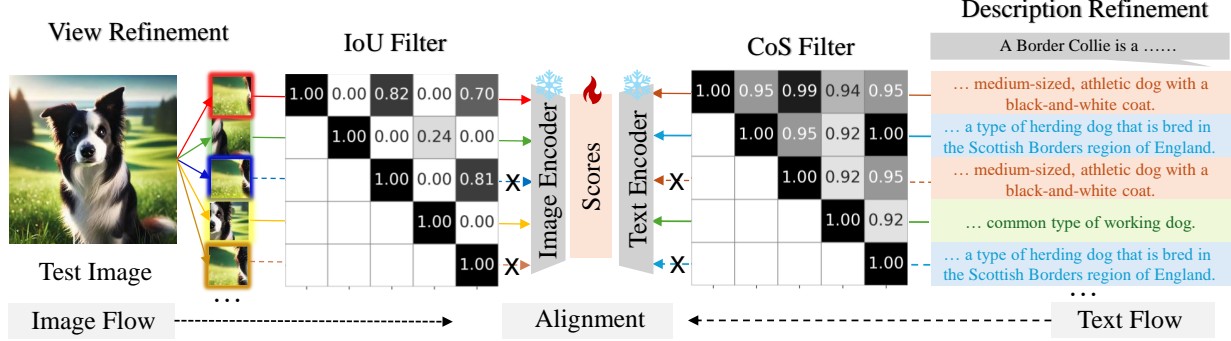

Figure 2: **An Overview of BiFTA.** To reduce potential redundancy in views and descriptions, the randomly cropped views undergo filtering with the IoU filter (Section 4.1), while the randomly sampled description texts are processed using the CoS filter (Section 4.2) when computing the similarity between a single image and a single label. The similarity score is then calculated on the refined views and descriptions.

where IoU(.) is the Jaccard index[1] (i.e., intersection over union) between two patches, and $1 - \delta$ is a hyperparameter representing the IoU threshold detailed in Section 5. When $f_{\mathrm{IoU}}(I_i, V) = 0$, the view set $V$ remains unchanged. However, if $f_{\mathrm{IoU}}(I_i, V) = 1$, $V$ will be appended with the new view $I_i$, i.e., $V \leftarrow V \cup \{I_i\}$. Thus, while keeping the size of $V$ unchanged, this effectively augments more effective views.

## 4.2 Description Refinement

In previous works, CuPL (Pratt et al., 2023) used label-integrated prompt templates such as "Describe what a/an [label] looks like" to obtain various appearance descriptions of different classes. Meanwhile, AttrVR (Cai et al., 2025) employs prompts like "Describe the appearance of [task] [label]" to obtain DesAttr (which describes intra-class features) and DistAttr (which distinguishes inter-class features). Here, we take the union of the descriptions obtained by these methods as the preliminary refinement data.

For a given label $y$, let us denote the three sets of descriptions as $T^{\mathrm{CuPL}}(y), T^{\mathrm{Des}}(y), T^{\mathrm{Dist}}(y)$, respectively. We can then formulate the equation as:

$$T^{\mathrm{CuPL}}(y) = f_{\mathrm{LLM}}(y|[\mathrm{cupl\_prompt}]), T^{\mathrm{Des}}(y) = f_{\mathrm{LLM}}(y|[\mathrm{des\_prompt}]), T^{\mathrm{Dist}}(y) = f_{\mathrm{LLM}}(y|[\mathrm{dist\_prompt}]),$$

where $f_{\mathrm{LLM}}(\cdot)$ returns the LLM output given queries, and $\mathrm{cupl\_prompt}, \mathrm{des\_prompt}, \mathrm{dist\_prompt}$ are aforementioned prompts used by these methods.

Similar to the *view refinement*, in order to alleviate the redundancy when obtaining description set $D = \{T_1, T_2, ..., T_m\}$, we also employ a filtering function $f_{\mathrm{CoS}}(\cdot, \cdot)$ during random sampling. When sampling text $T_i \in T^{\mathrm{CuPL}}(y) \cup T^{\mathrm{Des}}(y) \cup T^{\mathrm{Dist}}(y)$, the filter function $f_{\mathrm{CoS}}(T_i, D)$–which determines whether new sampled $T_i$ should be included in current set $D$–can be represented as $f_{\mathrm{CoS}}(T_i, D) = f_{\mathrm{CS}}(T_i, D) \cdot f_{\mathrm{TopK}}(T_i, D)$, which consists of a filter $f_{\mathrm{CS}}(\cdot, \cdot)$ that removes similar descriptions:

$$f_{\mathrm{CS}}(T_i, D) = \begin{cases} 1 & \forall T \in D, \cos(f_{\mathrm{txt}}(T), f_{\mathrm{txt}}(T_i)) < 1 - \epsilon \\ 0 & \exists T \in D, \cos(f_{\mathrm{txt}}(T), f_{\mathrm{txt}}(T_i)) \geq 1 - \epsilon \end{cases},$$

and another filter $f_{\mathrm{TopK}}(\cdot, \cdot)$ for eliminating noisy or irrelevant descriptions:

$$f_{\mathrm{TopK}}(T_i, D) = \begin{cases} 1 & T_i \in \mathrm{Top\text{-}K}(\cos(f_{\mathrm{txt}}(\hat{T}_y), f_{\mathrm{txt}}(T))|T \in D) \\ 0 & T_i \notin \mathrm{Top\text{-}K}(\cos(f_{\mathrm{txt}}(\hat{T}_y), f_{\mathrm{txt}}(T))|T \in D) \end{cases},$$

where $\epsilon$ is a hyperparameter representing the threshold, Top-K$(\cdot)$ returns the set of variable $T$s corresponding to the top $k$ function values, and $\hat{T}_y$ is the label prompt string (e.g. "This is a photo of a/an [label]").

---

[1] https://en.wikipedia.org/wiki/Jaccard_index

The use of $f_{\text{CS}}(\cdot, \cdot)$ ensures that the selected description set contains as little repetitive or redundant textual content as possible. The use of $f_{\text{TopK}}(\cdot, \cdot)$ minimizes the presence of distracting descriptions in the candidate description set (e.g., noisy text generated by an LLM). When $f_{\text{CoS}}(T_i, D) = 0$, the description set $D$ remains unchanged. However, if $f_{\text{CoS}}(T_i, D) = 1$, $D$ will be appended with the new description $T_i$, i.e., $D \leftarrow D \cup \{T_i\}$.

Overall, by applying these functions, the descriptions becomes more diverse under fixed set size $|D|$, effectively increasing the number of useful descriptions. This is equivalent to performing *description refinement*.

### 4.3 Overall Pipeline

The overall pipeline of BiFTA is illustrated in Figure 2. When computing the similarity between a image and a label, the randomly cropped views are filtered using the IoU filter $f_{\text{IoU}}$ and stored by a patch queu $V$, while the randomly sampled description texts are filtered using the CoS filter $f_{\text{CoS}}$. Then, the similarity between views and descriptions is computed using Eq. 1 to obtain the final prediction. The detailed algorithm is provided in Appendix A.

### 4.4 Understanding BiFTA Through Posterior Probability

In this section, we will explain that the label predictions obtained using our method contain more effective views and descriptions through the angle of posterior probability.

We will first define the concept of *Redundant Views/Descriptions* in Definition 1 and *Deduplicated Set* in Definition 2.

**Definition 1.** *(Redundant Views/Descriptions). Assuming $I_i$ and $I_j$ are two views of image $x$, if $\text{IoU}(I_i, I_j) \geq 1 - \delta$, where $1 - \delta$ is the IoU threshold, then $I_i$ and $I_j$ are considered to have a significant overlap, i.e., $I_i$ and $I_j$ are redundant views of each other. Regarding textual descriptions $T_p$ and $T_q$ for label $y$, if $\cos(f_{\text{txt}}(T_p), f_{\text{txt}}(T_q)) \geq 1 - \epsilon$, where $1 - \epsilon$ is the threshold, then the two descriptions can be considered nearly identical, i.e., they are mutually redundant descriptions.*

**Definition 2.** *(BiFTA-Deduplicated Set). For a set containing views $V = \{I_1, I_2, ..., I_n\}$ with the size $n$, $V$ is a deduplicated view set if and only if it satisfies:*

$$\forall I_i, I_j \in V, \text{IoU}(I_i, I_j) < 1 - \delta.$$

*Similarly, for a set $D = \{T_1, T_2, ..., T_m\}$ containing $m$ descriptions, $D$ is a deduplicated description set if and only if it satisfies:*

$$\forall T_p, T_q \in D, \cos(f_{\text{txt}}(T_p), f_{\text{txt}}(T_q)) < 1 - \epsilon.$$

Through Definition 2 and Section 4.1 to 4.3, it can be concluded that view and description sets used by our method are BiFTA-deduplicated set. However, view and description sets utilized in WCA do not belong to BiFTA-deduplicated sets, as no constraints are established during random cropping of images or sampling of description texts.

**Proposition 3.** *For a pair of redundant views $I_i$ and $I_j$ (cf. Definition 1), when $\delta \to 0$, then for a label $y$, it can be obtained that $p(I_i|y) \approx p(I_i, I_j|y)$. Similarly, for a pair of redundant descriptions $T_p$ and $T_q$, assuming the text encoder $f_{\text{txt}}(\cdot)$ is an injective function, when $\epsilon \to 0$, then $p(T_p|y) \approx p(T_p, T_q|y)$.*

Proposition 3 states that as $\delta$ approaches 0, the conditional probability of two mutually redundant views given the label $y$ approximately equals the conditional probability of one of the views given $y$. The redundancy descriptions follow accordingly. See detailed proof of Proposition 3 in Appendix B.1.

**Proposition 4.** *Assuming that the label $y \in \mathcal{Y}$ follows a uniform distribution, then applying our method, the posterior probability of $y$ can be formulated as:*

$$\hat{p}_{\text{ours}}(y|I_1, I_2, ..., I_n, T_1, T_2, ..., T_m) \propto p(I_1, I_2, ..., I_n, T_1, T_2, ..., T_m|y),$$

*where both $\{I_1, I_2, ..., I_n\}$ and $\{T_1, T_2, ..., T_m\}$ are BiFTA-deduplicated sets (cf.Definition 2). Under the same condition in Proposition 3, when applying WCA, the posterior probability can be estimated as:*

$$\hat{p}_{\text{wca}}(y|I_1', I_2', ..., I_n', T_1', T_2', ..., T_m') \propto p(V', D'|y),$$

Table 1: Zero-shot classification accuracy (%) across various baseline methods with the pre-trained CLIP model (ViT-B/32). We report the averaged results and standard deviations $\sigma$ of 20 runs, with the improvement $\Delta$(%) over the top-performing baseline WCA highlighted in **green**. The results of our method are highlighted and we use **bold** to represent the best-performing method.

| Method | ImageNet | CUB | Oxford Pets | DTD | Food101 | Place365 |
|---|---|---|---|---|---|---|
| CLIP | 62.05 | 51.21 | 85.04 | 42.93 | 82.60 | 38.51 |
| CLIP-E | 63.37 | 52.74 | 87.38 | 43.83 | 83.93 | 39.28 |
| CLIP-D | 63.01 | 52.69 | 84.46 | 44.20 | 84.12 | 39.90 |
| Waffle | 63.30 | 52.04 | 85.50 | 42.98 | 83.98 | 39.47 |
| CuPL | 64.37 | 49.76 | 87.03 | 47.50 | 84.20 | 39.08 |
| WCA | 66.49 | 56.74 | 89.05 | 49.89 | 86.11 | 40.55 |
| **BiFTA (ours)** | **66.58±0.06** | **58.24±0.17** | **89.74±0.14** | **53.22±0.26** | **86.43±0.06** | **41.55±0.06** |
| $\Delta$ | **+0.09** | **+1.50** | **+0.69** | **+3.33** | **+0.32** | **+1.00** |

Table 2: Average classification accuracy (%) across various baseline methods with different CLIP models. The improvements $\Delta$(%) over the top-performing baseline (i.e., WCA) are highlighted in **green**. We use **bold** to represent the best-performing method and underlined to represent the second-best method.

| Model Architecture | CLIP | CLIP-E | CLIP-D | Waffle | CuPL | WCA | BiFTA (ours) | $\Delta$ |
|---|---|---|---|---|---|---|---|---|
| ViT-B/32 | 60.39 | 61.76 | 61.40 | 61.21 | 62.16 | 64.81 | **65.96** | +1.15 |
| ViT-B/16 | 63.59 | 64.51 | 64.67 | 64.34 | 66.09 | 67.87 | **68.29** | +0.42 |
| ViT-L/14 | 68.94 | 70.12 | 69.87 | 69.61 | 71.31 | 72.50 | **72.98** | +0.48 |
| RN-50 | 56.97 | 58.64 | 58.39 | 57.92 | 60.01 | 62.00 | **62.54** | +0.54 |
| RN-101 | 59.14 | 60.50 | 59.22 | 58.89 | 59.04 | 61.14 | **62.03** | +0.89 |

*where $V'$ and $D'$ are the largest deduplicated set satisfying $V' \subseteq \{I_1', I_2', ..., I_n'\}, D' \subseteq \{T_1', T_2', ..., T_m'\}$. The actual effective numbers of views and descriptions should be less than or equal to that of our method (i.e., $|V'| \leq m, |D'| \leq n$).*

According to Proposition 4 (proved in Appendix B.2), when the numbers of views and descriptions are fixed, our refinement-and-filtering approach prioritizes the more effective ones while filtering out redundant information. This is equivalent to increasing the number of effective views and descriptions after removing duplicated ones.

## 5 Experiment

In this section, we first provide the detailed experimental settings in Section 5.1. We evaluate the effectiveness of BiFTA through comprehensive experiments across 6 benchmark datasets and 5 different CLIP model architectures. In Section 5.2, we present the main experimental result of utilizing the CLIP model with a ViT-B/32 backbone across various datasets. Additionally, we provide complete experimental results of other model architectures in Appendix C. In Section 5.3, we conduct ablation studies to validate our design principles of *view refinement* and *description refinement*. Limitations are presented in Appendix F.

### 5.1 Experimental Settings

**Datasets.** To evaluate BiFTA, we conduct experiments on 6 downstream classification tasks under a zero-shot setting, including: (1) ImageNet (Deng et al., 2009), a large-scale dataset comprising 1,000 diverse object classes; (2) CUB (Welinder et al., 2010), a fine-grained dataset of 200 bird species, focusing on subtle visual distinctions; (3) Oxford Pets (Parkhi et al., 2012), a dataset of 37 pet categories; (4) DTD (Cimpoi et al., 2014), a texture dataset containing 47 categories of materials and surfaces; (5) Food101 (Bossard et al., 2014),

Figure 3: A visualization comparing the effectiveness of *view refinement* and *description refinement* in BiFTA against WCA. **Left**: with an IoU filter, the cropped samples exhibit diverse and distinctive localized features. **Right**: with a CoS filter, the texts can describe various local features of a category.

a dataset of 101 food categories; and (6) Place365 (Zhou et al., 2017), a scene recognition dataset with 365 categories of indoor and outdoor environments. These datasets span a wide range of categories, encompassing various visual domains, involving scenes, textures, food, animals and fine-grained objects. This ensures the robustness and generalizability of BiFTA across various real-world applications.

**Baselines.** Our evaluation of zero-shot visual classification integrates 6 baselines: (1) CLIP (Radford et al., 2021), a naive approach that incorporates a manually crafted label prompt; (2) Ensemble CLIP (CLIP-E) (Radford et al., 2021), an advance approach that incorporates a series of label prompts; (3) CLIP-D (Menon & Vondrick, 2023), an approach that utilizes category descriptions generated by a LLM instead of label prompting approach; (4) Waffle (Roth et al., 2023), a novel approach that replaces LLM-generated category descriptions with random word descriptions; (5) CuPL (Pratt et al., 2023), a method that leverages LLM and improves the quality and variety of textual descriptions compared with CLIP-D; (6) WCA (Li et al., 2024), a method that cross-aligns localized image patches with fine-grained textual descriptions.

**Implementation Details.** We use CLIP as the pre-trained model and evaluated on both *Vision Transformer* (ViT) and ResNet backbone architectures, specifically ViT-B/32, ViT-B/16, ViT-L/14, RN-50 and RN-101. These architectures are selected to enable a thorough analysis of the proposed method across varying scales and complexities. We keep the shared hyperparameters consistent with WCA settings (Li et al., 2024): we use $n = 60$ for the patch queue length (this is equivalent to number of crops) and $m = 50$ for the number of textual descriptions per category. We utilize the same cropping window size ranging from $[\alpha_{\mathbf{low}}, \beta_{\mathbf{high}}]$, where $\alpha_{\mathbf{low}} = 0.5$ and $\beta_{\mathbf{high}} = 0.9$ across all experiments. Additionally, we adopt the same technique as WCA to store the embeddings of localized image patches during the initial execution (Li et al., 2024). This allows for their reuse when evaluating different sets of textual descriptions, significantly reducing computational costs.

## 5.2 Zero-shot Image Classification Results

In Table 1, the classification performance of CLIP (B/32) underscores the consistent superiority of BiFTA over existing baselines. In specific, BiFTA achieves an average performance gain of $\Delta_{\mathbf{avg}} = \mathbf{1.15}\%$ compared to WCA across all downstream tasks and outperforms previous baselines that utilize label prompts by a large margin. Notably, BiFTA achieves a **3.33**% accuracy gain over WCA on the DTD dataset, which contains various texture information. This improvement stems from the dataset's inherent challenges: textual patterns in DTD are often homogeneous, and random cropping tends to produce semantically similar image regions. To address this, our *view refinement* systematically prunes similar image patches using the IoU thresholds.

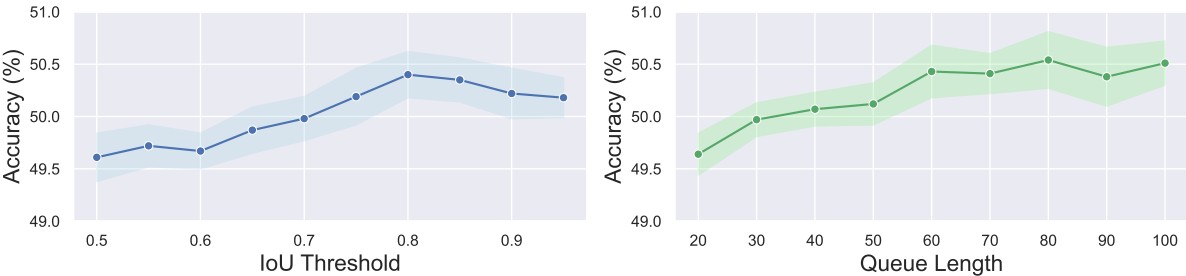

Figure 4: **Left**: Accuracy of different IoU thresholds on DTD dataset with CLIP (B/32). **Right**: Accuracy of increasing the patch queue length of storing cropped samples on DTD dataset with CLIP (B/32).

The resulting patch queue $V$ consists of semantically independent candidates. Then, they are dynamically weighted via a softmax function shown in Section 3, our visual refinement can effectively adjust a proper weight to each image patch from the process. This allows the model to prioritize discriminative patches during inference, enhancing robustness to repetitive textures. Furthermore, in Table 2, BiFTA exhibit similar trend observed in WCA that smaller backbone models (e.g., B/32) exhibit more significant improvements ($+1.15\%$) comparing with their larger counterparts (e.g., L/14), ($+0.48\%$). This consistency confirms that our refinements retain the visual-text cross-alignment principles of WCA. More experimental results on other CLIP model architectures can be found in Appendix C. The average results over all benchmarks are concluded in Table 2, where we show that BiFTA achieves the best performance within various model architectures.

In Figure 3, the left section shows an image of a goose, BiFTA captures semantically diverse image regions (e.g., head, wings, neck and fur), whereas WCA exhibit redundancy in acquired image patch (e.g., repeated neck or irrelevant patches). This demonstrates how *view refinement* mitigates feature overlapping through IoU-guided filtering. The right section presents some text samples describing a goose, we see that sample descriptions through CoS Filter are more diverse and distinctive. The CoS Filter (Section 4.2) retains semantically distinct prompts (e.g., "plump, elongated body." and "a long, slender body"), while pruning redundant phrases (e.g., "long necks and webbed feet").

### 5.3 Hyperparameter Analysis and Ablation Studies

**Hyperparameter tuning.** Figure 4 illustrates the impact of the hyper-parameters: IoU threshold $\eta = 1 - \delta$ and patch queue length $L = |V|$. The experiments are conducted on the DTD dataset using CLIP (B/32). On the left, the classification accuracy varies as the $\eta$ changes, with accuracy gradually rising and then dropping as $\eta$ increases, which suggest that a moderate $\eta$ is needed to achieve an optimal performance. In practice, a lower $\eta$ often results in a critic threshold where insufficient image patches are satisfied to build up the patch queue. To address this, we re-sample from the patch queue until the required length $L$ is met, this could potentially lead to more redundant image patches in the patch queue. The trend further suggests that the redundant image patches negatively influence the classification accuracy. Notably, BiFTA performance converges to WCA as $\eta$ approaches 1, where the decreasing trend since IoU = 0.80 indicates the *view refinement* is effective. For consistency across all experiments, we set $\eta = 0.80$. On the right, it shows that increasing the patch queue length $L$ (equivalent to increase patch samples) has a positive impact on classification accuracy, whereas the increasing trend reaches a plateau at $L = 60$.

**Ablation studies of various description refinement.** Figure 5 presents the ablation studies of choosing different set of textual descriptions. These textual descriptions are generated by LLM with different prompt template designs, which are introduced in Section 4.2. Only the mixed description set from our *description refinement* shown superior performance over most of the downstream tasks, where the shown result is averaged across 3 different CLIP architectures: ViT-B/32, ViT-B/16 and ViT-L/14. Also, the dash line exhibits the average results of WCA method, we compare and observe that BiFTA can show different extent of improvement to WCA with various textual description sets. Notably, textual descriptions generated by RAG-prompt templates are unsatisfactory comparing with other description sets, we include details of the implementation of RAG-prompt templates in Appendix D. In addition, the performance of CuPL shown

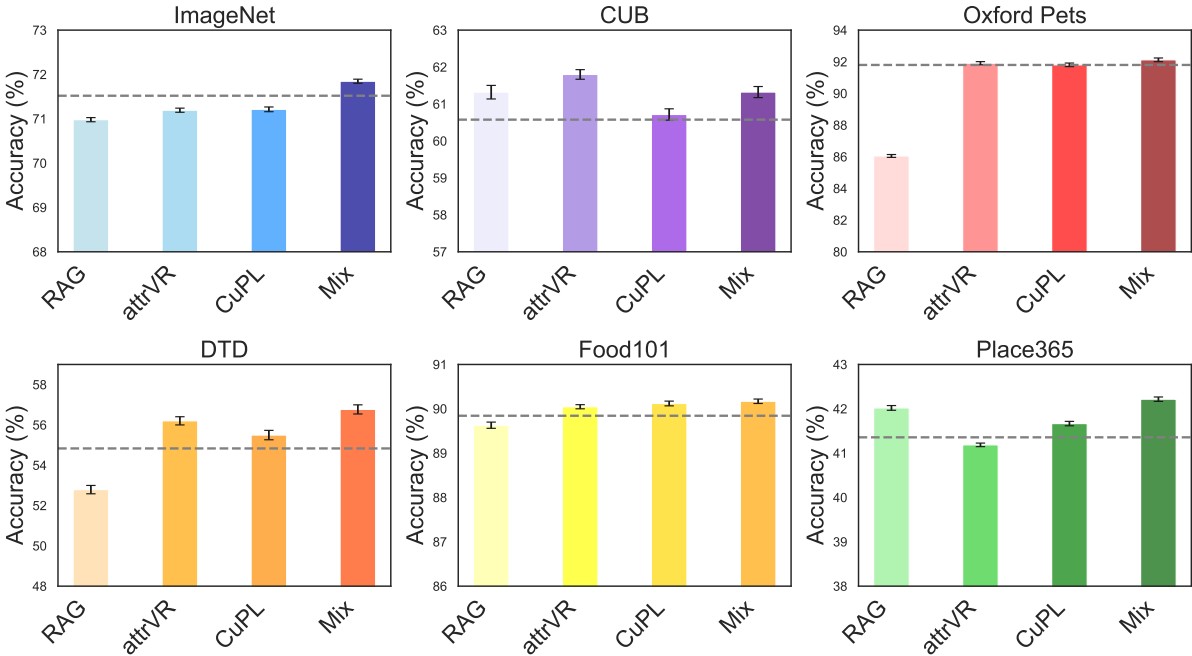

Figure 5: Comparing *view refinement* with different description sets and strategies on BiFTA. The results are averaged across 3 image encoder backbones of the CLIP model, where 'mix' is the strategy BiFTA finally utilized for *view refinement*.

in the chart is equivalent to only adapt *view refinement* of BiFTA. This suggests that only adapting *view refinement* is enough to boost the zero-shot image classification performance. There are more detailed results shown in Appendix E. Furthermore, we also evaluate the changes of $\beta$ and observed that larger $\beta$ yields better results as demonstrated in Table 7 of Appendix E.

**Ablation studies of BiFTA w/o view refinement and description refinement.** We also compare the experimental results of WCA with a complete version of BiFTA and partial BiFTA in Table 8 of Appendix E. We show that BiFTA with refinement on single modality is enough to improve the cross-alignment performance. For ImageNet and Food101 datasets, the models often exhibit better performance with BiFTA w/o description refinements, which indicates our merged description set might not be an optimal description set. As we inspired from the RAG-prompt template discussed in Appendix D, we believe that utilizing a larger database could improve the quality of textual descriptions. Hence, the description set could be further improved by integrating RAG with a considerable size database. Overall, BiFTA with both modality refinements achieves the best average results across all three CLIP model architectures.

## 6   Conclusion

In this work, we identified a critical limitation in existing fine-grained visual-text alignment methods: the presence of redundant information in both localized image patches and LLM-generated textual descriptions. To address this, we propose BiFTA, a novel framework that introduces two key innovations: (1) *view refinement* via IoU-based filtering to eliminate spatially overlapping image patches, and (2) *description refinement* through cosine similarity thresholding to remove semantically redundant textual descriptions. Our experiments across 6 benchmark datasets demonstrate that BiFTA consistently outperforms state-of-the-art methods in zero-shot classification accuracy over the previous methods. The ablation studies validate the necessity of both components: IoU filtering ensures diverse visual features, while cosine-based text pruning enhances semantic specificity. Theoretical analysis further justifies our method by linking redundancy reduction to improved posterior probability estimation in multi-modal alignment.

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

# A   Appendix 1: Additional Algorithms for the BiFTA Implementation

---
**Algorithm 1** Zero-Shot Classification Pipeline of BiFTA
---
1: **Input** Query image $I_0 \in \mathbb{R}^{H \times W \times 3}$; labels $y \in \mathcal{Y}$; hyperparameters: a patch queue $Q$, crop count $N$, textual description count $M$, a label prompt $\hat{T}_y$, and pre-trained CLIP model with encoders $f_{\text{img}}(\cdot)$ (image) and $f_{\text{txt}}(\cdot)$ (text).
2: **Initialize** $Q \leftarrow [I_1]$ by Eq. 2
   \# Step 1: View Refinement
3: **for** $i = 2$ to $N$ **do**
4:     **Generate** $I_i$ by Eq. 2
5:     **Check Is\_Redundant**$(I_i)$ by Algo. 2
6:     **Is\_Redundant**$(I_i) ==$ **false**$, Q.$push$(I_i)$
7:     **Compute** $w_i = \text{softmax}(\cos(f_{\text{img}}(I_0), f_{\text{img}}(I_i)))$
8: **end for**
   \# Step 2: Description Refinement
9: **for** $y \in \mathcal{Y}$ **do**
10:     **Obtain** $\mathcal{T}^y = \{T_j\}_{j=1}^J$ where $J > M$
11:     **Remove** $T_j \in \mathcal{T}^y$ based on Algo. 3
12:     **Obtain** $\tilde{\mathcal{T}}^y \subseteq \mathcal{T}^y$ from Line (11)
13:     **Initialize** $\hat{T}_y$
14:     **Select top-**$M$ $T$s by $\arg\max_{T \in \tilde{\mathcal{T}}^y} \cos(f_{\text{txt}}(T), f_{\text{txt}}(\hat{T}_y))$
15:     **Compute** $v_j$ for all $\{T_j\}_{j=1}^M$ via $v_j = \text{softmax}(\cos(f_{\text{txt}}(\hat{T}_y), f_{\text{txt}}(T_j))$
16:     **Compute** $\text{sim}_{\text{WCA}}^y$ via Eq. 1
17: **end for**
18: **Output** $y^* = \arg\max_{y \in \mathcal{Y}} \text{sim}_{\text{WCA}}^y$

---
**Algorithm 2** Implementation of Redundant Image Patch Filtering
---
1: **Input:** A patch queue $Q$ contains all the previous saved image patches, a new cropped image patch $I_i$, an IoU threshold $\eta = 1 - \delta$.
2: **Initialize Is\_Redundant** $==$ **false**
3: **for** $k = 1$ to $|Q|$ **do**
4:     **Compute** $\eta_k = \text{IoU}(I_i, Q[k])$
5:     **if** $\eta_k \geq \eta$ **then**
6:         **Is\_Redundant** $\leftarrow$ **true**, **break**
7:     **end if**
8: **end for**
9: **Output: Is\_Redundant**

---
**Algorithm 3** Implementation of Redundant Textual Description Filtering
---
1: **Input:** A set of textual descriptions of label $y$, $\mathcal{T}^y = \{T_j\}_{j=1}^J$, where $J$ is the number of descriptions in the merged textual description set; $\tau = 1 - \epsilon$ is the tolerance for the duplicate texts, setting to 1.0, $f_{\text{txt}}$ is a text encoder.
2: **for** $j = 1$ to $J - 1$ **do**
3:     **for** $k = j + 1$ to $J$ **do**
4:         **Compute** $S = \cos(f_{\text{txt}}(T_j), f_{\text{txt}}(T_k))$
5:         **if** $S \geq \tau$ **then**
6:             **Remove** $T_k$ from $\mathcal{T}^y$
7:         **end if**
8:     **end for**
9: **end for**
10: **Obtain** $\tilde{\mathcal{T}}^y = \mathcal{T}^y$
11: **Output:** $\tilde{\mathcal{T}}^y$

# B    Appendix 2: Detailed Proof

## B.1    Proof for Proposition 3

*Proof.*

**Redundant Views:** For views variables $I_i$ and $I_j$, when $\delta \to 0$, we have $\text{IoU}(I_i, I_j) \to 1$. It means that two views are nearly completely overlapping, which is equivalent to:

$$||I_i - I_j||_2 \to 0,$$

where $|| \cdot ||_2$ represents the Euclidean norm.,

By applying the conditional probability formula, it is not difficult to obtain:

$$p(I_i|y) \cdot p(I_j|I_i, y) = p(I_i, I_j|y),$$

where:

$$p(I_j|I_i, y) = \frac{p(I_j, I_i, y)}{p(I_i, y)}.$$

As $I_i$ nearly completely overlaps with $I_j$, we got: $p(I_i, y) \to p(I_j, I_i, y)$, and thus:

$$p(I_j|I_i, y) \to 1.$$

Conclusively, we have:

$$\lim_{\text{IoU}(I_i, I_j) \to 1} p(I_i|y) \to p(I_i, I_j|y).$$

Therefore, when $\delta \to 0$, we have $p(I_i|y) \approx p(I_i, I_j|y)$.

**Redundant Descriptions:** For description variable $T_p$ and $T_q$, when $\epsilon \to 0$, we have $\cos(f_{\text{txt}}(T_p), f_{\text{txt}}(T_q)) \to 1$. It implies that the text embeddings of these two descriptions are nearly identical, which is equivalent to:

$$||f_{\text{txt}}(T_p) - f_{\text{txt}}(T_q)||_2 \to 0.$$

Similarly, following the same steps above, we will have:

$$\lim_{\cos(f_{\text{txt}}(T_p), f_{\text{txt}}(T_q)) \to 1} p(T_p|y) \to p(T_p, T_q|y).$$

Therefore, when $\epsilon \to 0$, assuming the fixed text encoder $f_{\text{txt}}(\cdot)$ is an injective function, we have $p(T_p|y) \approx p(T_p, T_q|y)$.

$\square$

## B.2    Proof for Proposition 4

According to Bayes' theorem, we can obtain

$$p(y|I_1, I_2, ..., T_n, T_1, T_2, ..., T_m) = \frac{p(I_1, I_2, ..., I_n, T_1, T_2, ..., T_m|y) \cdot p(y)}{p(I_1, I_2, ..., T_n, T_1, T_2, ..., T_m)}.$$

As is assumed, the downstream label $y \in \mathcal{Y}$ follows the uniform distribution. Therefore, the prior probability satisfies $p(y_i) = p(y_j), \forall y_i, y_j \in \mathcal{Y}$. Besides, $p(I_1, I_2, ..., T_n, T_1, T_2, ..., T_m)$ can be regarded as the normalization factor. Therefore, we have:

$$p(y|I_1, I_2, ..., T_n, T_1, T_2, ..., T_m) \propto p(I_1, I_2, ..., I_n, T_1, T_2, ..., T_m|y).$$

Applying $V, D$ to represent the sets, we have $V = \{I_1, I_2, ..., I_n\}, D = \{T_1, T_2, ..., T_m\}$. We assume that $V' \in V, D' \in D$ represent the largest BiFTA-deduplicated sets of $V, D$. Then:

$$\forall I_i \in V - V' : \exists I_j \in V', \mathrm{IoU}(I_i, I_j) \geq 1 - \delta, \text{and}$$

$$\forall I_i, I_j \in V, \mathrm{IoU}(I_i, I_j) < 1 - \delta.$$

Similarly, for textual descriptions, we have:

$$\forall T_p \in D - D' : \exists T_q \in D', \cos(f_{\mathrm{txt}}(I_i), f_{\mathrm{txt}}(I_j)) \geq 1 - \epsilon, \text{and}$$

$$\forall T_p, T_q \in D, \cos(f_{\mathrm{txt}}(I_i), f_{\mathrm{txt}}(I_j)) < 1 - \epsilon.$$

For any $I_i$ in $V - V'$, there exists an element $I_j \in V'$ such that $I_i$ and $I_j$ are mutually redundant. The same applies to the description set. Therefore, according to Proposition 3, we can obtain

$$p(y|I_1, I_2, ..., T_n, T_1, T_2, ..., T_m) \propto p(V, D|y) \approx p(V', D'|y).$$

Since the view sets and description sets obtained through our method are BiFTA-deduplicated sets, while the sets obtained by WCA are not BiFTA-deduplicated sets, the statement in Proposition 4 can be derived.

## C  Appendix 3: More Experimental Results

Table 3: Zero-shot classification accuracy (%) across various baseline methods with the pre-trained CLIP model (ViT-B/16). We report the averaged results and standard deviations $\sigma$ of 20 runs, with the improvement $\Delta(\%)$ over the top-performing baseline WCA highlighted in **green** and **red**. The results of our method are highlighted and we use **bold** to represent the best-performing method.

| Method | ImageNet | CUB | Oxford Pets | DTD | Food101 | Place365 |
|---|---|---|---|---|---|---|
| CLIP | 66.74 | 56.01 | 88.14 | 42.98 | 88.40 | 39.27 |
| CLIP-E | 68.37 | 56.16 | 89.10 | 45.27 | 88.83 | 40.30 |
| CLIP-D | 68.04 | 57.08 | 87.52 | 46.17 | 88.85 | 40.34 |
| Waffle | 68.12 | 56.89 | 86.51 | 44.68 | 89.06 | 40.76 |
| CuPL | 69.61 | 56.42 | 91.14 | 50.53 | 88.98 | 39.83 |
| WCA | 71.05 | 59.87 | 92.13 | 52.87 | 89.99 | 41.33 |
| BiFTA (ours) | **71.14±0.04** | **60.06±0.15** | **91.67±0.11** | **54.64±0.16** | **90.11±0.05** | **42.12±0.04** |
| $\Delta$ | **+0.09** | **+0.19** | **-0.46** | **+1.77** | **+0.12** | **+0.79** |

Table 4: Zero-shot classification accuracy (%) across various baseline methods with the pre-trained CLIP model (ViT-L/14). We report the averaged results and standard deviations $\sigma$ of 20 runs, with the improvement $\Delta(\%)$ over the top-performing baseline WCA highlighted in **green**. The results of our method are highlighted and we use **bold** to represent the best-performing method.

| Method | ImageNet | CUB | Oxford Pets | DTD | Food101 | Place365 |
|---|---|---|---|---|---|---|
| CLIP | 73.48 | 62.12 | 93.24 | 52.61 | 92.55 | 39.63 |
| CLIP-E | 75.52 | 62.53 | 93.62 | 55.43 | 93.07 | 40.55 |
| CLIP-D | 75.03 | 63.26 | 93.30 | 55.05 | 93.03 | 40.55 |
| Waffle | 75.31 | 62.27 | 91.55 | 54.31 | 93.33 | 40.89 |
| CuPL | 76.62 | 62.15 | 94.33 | 60.59 | 93.37 | 40.77 |
| WCA | 77.32 | 65.12 | 94.67 | 61.74 | 93.93 | 42.19 |
| BiFTA (ours) | **77.82±0.04** | **65.67±0.13** | **94.96±0.10** | **62.45±0.26** | **93.97±0.04** | **42.98±0.05** |
| $\Delta$ | **+0.50** | **+0.55** | **+0.29** | **+0.71** | **+0.04** | **+0.79** |

Table 5: Zero-shot classification accuracy (%) across various baseline methods with the ResNet-based CLIP model (RN-50). We report the averaged results and standard deviations $\sigma$ of 20 runs, with the improvement $\Delta(\%)$ over the top-performing baseline WCA highlighted in **green**. The results of our method are highlighted and we use **bold** to represent the best-performing method.

| Method | ImageNet | CUB | Oxford Pets | DTD | Food101 | Place365 |
|---|---|---|---|---|---|---|
| CLIP | 58.15 | 45.67 | 83.65 | 38.67 | 78.62 | 37.04 |
| CLIP-E | 59.82 | 46.58 | 85.66 | 41.22 | 80.82 | 37.73 |
| CLIP-D | 59.62 | 47.76 | 83.70 | 42.23 | 79.92 | 37.13 |
| Waffle | 59.82 | 46.76 | 83.54 | 38.88 | 80.74 | 37.77 |
| CuPL | 61.43 | 47.91 | 87.05 | 47.39 | 80.50 | 37.78 |
| WCA | 62.82 | 50.16 | 88.40 | 49.45 | 81.25 | 38.91 |
| BiFTA (ours) | **63.54±0.06** | **50.43±0.17** | **88.74±0.12** | **51.41±0.22** | **81.39±0.09** | **39.70±0.06** |
| $\Delta$ | **+0.72** | **+0.27** | **+0.34** | **+1.96** | **+0.14** | **+0.79** |

Table 6: Zero-shot classification accuracy (%) across various baseline methods with the ResNet-based CLIP model (RN-101). We report the averaged results and standard deviations $\sigma$ of 20 runs, with the improvement $\Delta(\%)$ over the top-performing baseline WCA highlighted in **green** and **red**. The results of our method are highlighted and we use **bold** to represent the best-performing method.

| Method | ImageNet | CUB | Oxford Pets | DTD | Food101 | Place365 |
|---|---|---|---|---|---|---|
| CLIP | 61.26 | 49.34 | 84.96 | 40.05 | 82.44 | 36.77 |
| CLIP-E | 62.31 | 49.65 | 86.97 | 43.62 | 83.64 | 37.81 |
| CLIP-D | 60.65 | 50.29 | 82.53 | 42.82 | 83.25 | 35.75 |
| Waffle | 61.25 | 48.05 | 83.70 | 40.05 | 82.48 | 37.83 |
| CuPL | 61.43 | 42.85 | 87.63 | 43.83 | 82.74 | 35.77 |
| WCA | 62.82 | 44.64 | 87.43 | 49.91 | 83.92 | 38.11 |
| BiFTA (ours) | **65.24±0.05** | **45.86±0.12** | **86.81±0.15** | **50.87±0.23** | **84.02±0.05** | **39.40±0.06** |
| $\Delta$ | **+2.42** | **+1.22** | **-0.62** | **+0.96** | **+0.10** | **+1.29** |

Tables 3 - 6 present the results of methods using various architectures of the CLIP image encoder. For all models, we employed the same experimental settings as those used in CLIP (B/32). The results demonstrate significant improvements on the DTD dataset, which are consistent across all CLIP architectures. In specific, BiFTA achieves an average accuracy improvement of 0.42% to 0.90% across 6 benchmarks compared to WCA. Notably, CLIP models with ResNet-based backbones exhibit larger performance improvements compared to other architectures. In summary, BiFTA consistently outperforms other baselines across all benchmarks, with the exception of CLIP (B/16) on the Oxford Pets dataset.

# D  Appendix 4: RAG-based Text Generation

We further investigate a novel prompt template based on *Retrieval-Augmented Generation (RAG)*, but the performance is unexpected as shown in figure 5. First, we begin by constructing a knowledge database using a pre-processed Wikipedia dataset [2] comprising 1.8 million documents. These document data are truncated, tokenized and encoded into embedding vectors using the "text-embedding-ada-002" text encoding model[3], which are then efficiently stored in the ChromaDB vector database. To generate textual descriptions, we manually craft a prompt template that integrates user queries with retrieved documents. The retrieval process relies on semantic similarity between the query and the database entries. These prompts are subsequently fed into a GPT model to produce the final textual descriptions. Despite these efforts, the resulting descriptions perform poorly when applied to CLIP-based classification tasks. Notably, we only utilize a small subset

---

[2] https://huggingface.co/datasets/Salesforce/wikitext/viewer/wikitext-103-v1
[3] An embedding model published by OpenAI.

(approximately 150k documents) of the whole dataset, as it would take approximately 46 days to save those encoded documents into the database. Hence, we hypothesize that the retrieved documents may lack relevance to the target categories. We leave it as future work to explore whether expanding the database size could improve retrieval accuracy and then enhance the textual descriptions.

## E   Appendix 5: Extra Ablation Studies

Table 7: The ablation study on $\beta$. This experiment is evaluated on the ImageNet dataset by leveraging CLIP (B/32). We set the lower bound $\alpha$ to 0.5 and report the Top-1 Accuracy (%).

| $\alpha = 0.5$ | $\beta$ | | | | |
|---|---|---|---|---|---|
| | 0.6 | 0.7 | 0.8 | 0.9 | 1 |
| WCA | 61.77±0.06 | 63.21±0.05 | 64.45±0.05 | 66.49±0.07 | 66.06±0.08 |
| Ours | 64.40±0.07 | 65.46±0.07 | 65.91±0.05 | 66.58±0.06 | 66.73±0.08 |

In Table 7, the ablation study compares the Top-1 accuracy (%) of WCA and BiFTA when varying the window size upper bound $\beta$. We observe a consistent growing trend of accuracies as the $\beta$ increases, where BiFTA exhibits a significant improvement when $\beta$ is low. This suggests that when the cropping windows are smaller, redundant small patches have a more pronounced negative impact on the weighted scores, whereas this issue is effectively addressed by integrating the *view refinement* introduced in BiFTA.

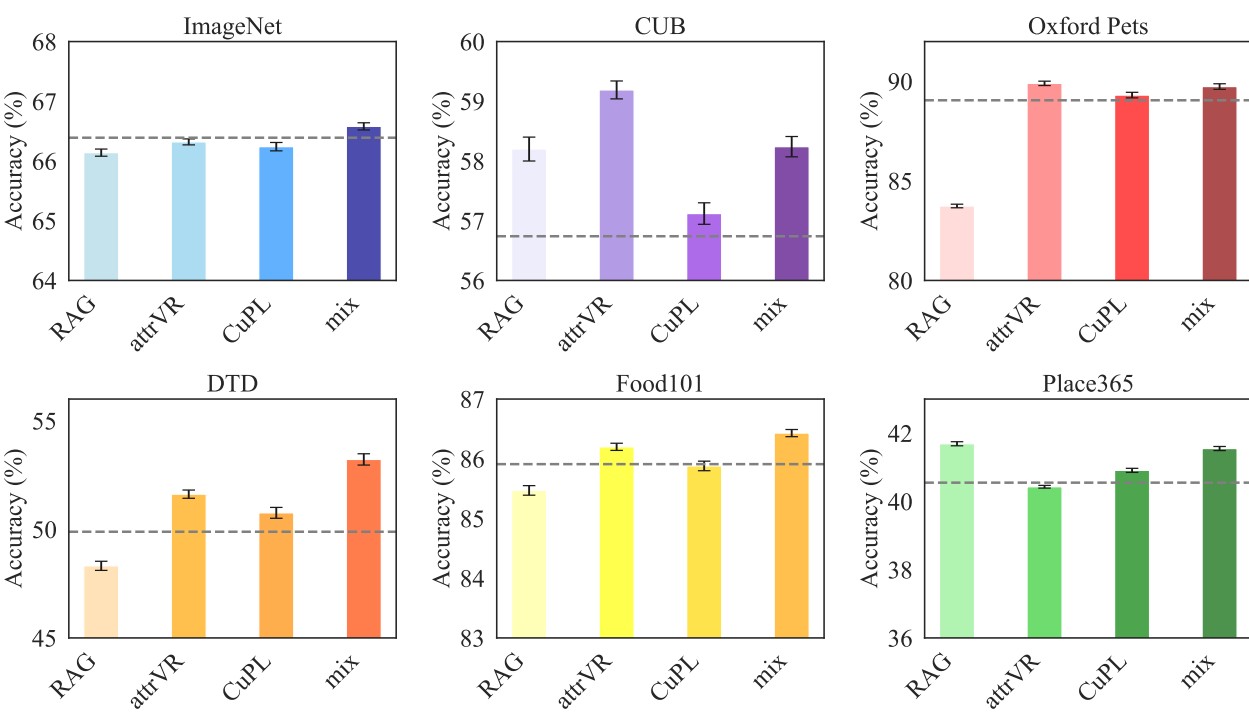

Figure 6: Results of exploring textual description studies, using the results of CLIP (B/32). "RAG" computes similarity using RAG-generated descriptions. "attrVR" uses both descriptive and distinctive texts to calculate similarity. "CuPL" directly employs descriptions from the CuPL method. "mix" combines "attrVR" and "CuPL" descriptions.

We provide additional experimental results for ablation studies by selecting different sets of textual descriptions. Figures 6 to 8 present the classification accuracy across all benchmarks for the 4 different description sets. Each figure represents the results obtained from CLIP ViT-B/32, ViT-B/16 and ViT-L/14, respectively. In

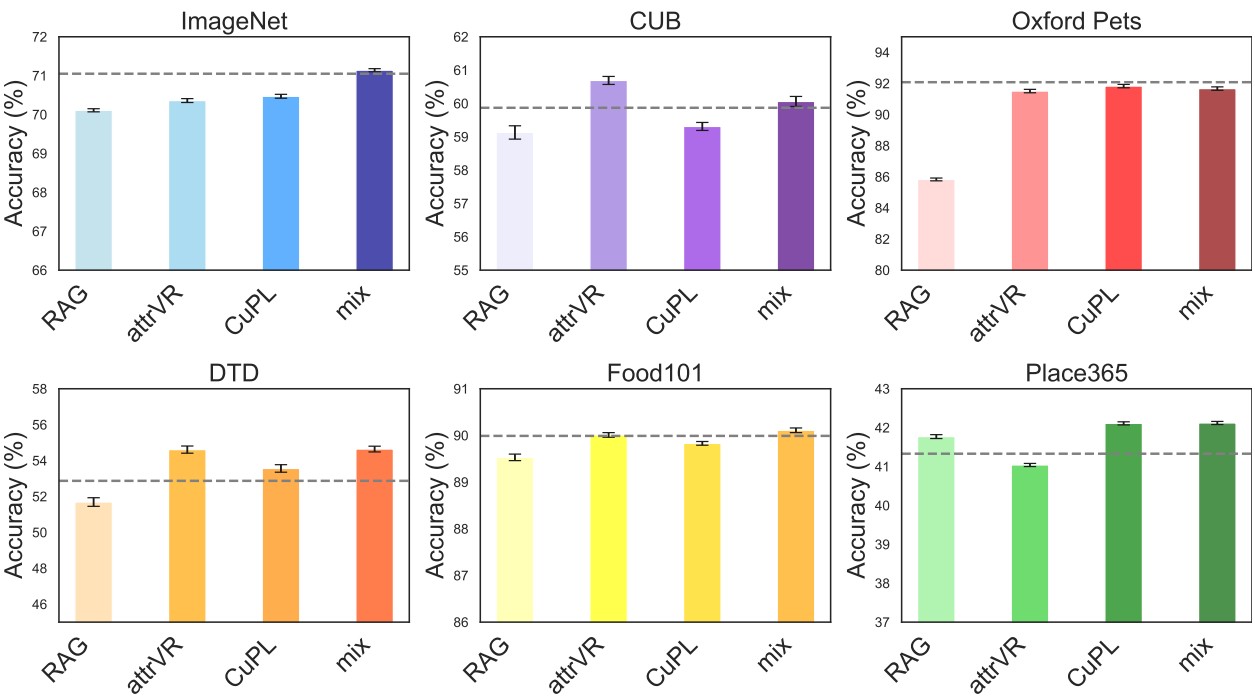

Figure 7: Results of exploring textual description studies, using the results of CLIP (B/16).

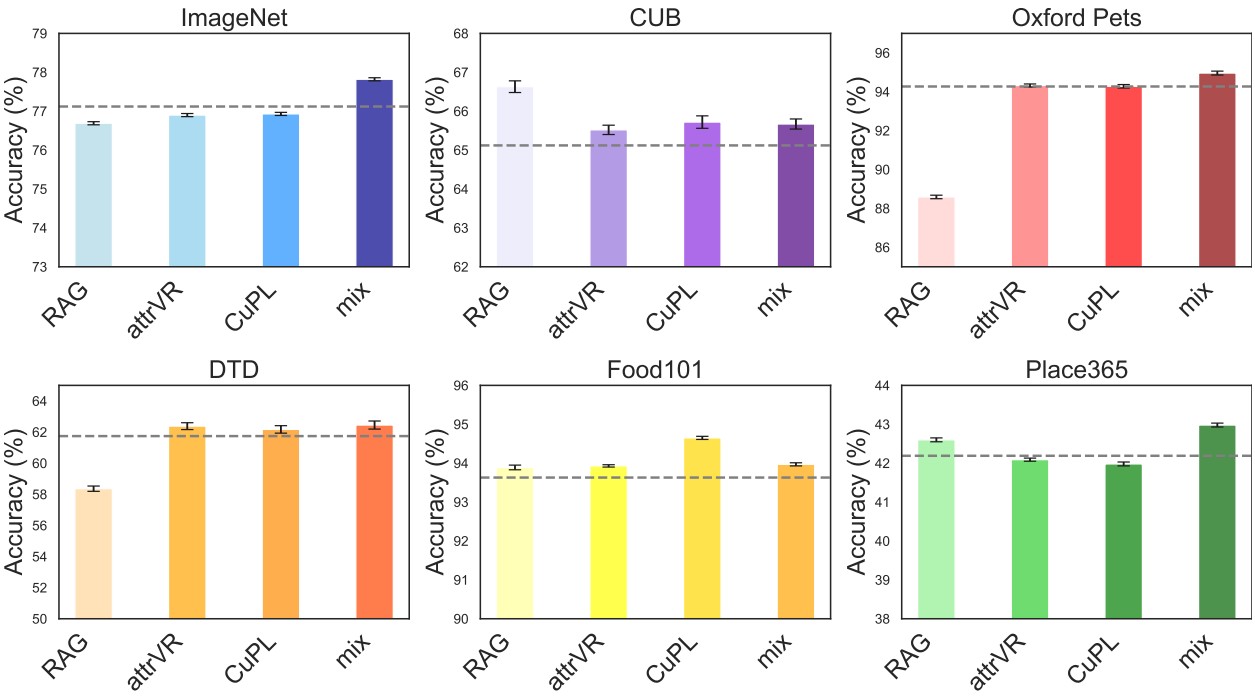

Figure 8: Results of exploring textual description studies, using the results of CLIP (L/14).

conclusion, the mixed set of descriptions, combining the CuPL and AttrVR descriptions and filtering them based on top-k similarities, demonstrates superior performance compared to other description sets.

Table 8: Ablation studies of comparing the performance of WCA and BiFTA between single modality refinement (w/o VF and w/o DF) and full refinements, using CLIP models (B/32, B/16 and L/14). **VF**: View Refinement; **DF**: Description Refinement. The best result for a single dataset across each model is underlined, and the best averaged results (%) are highlighted in **bold**.

|  | Methods | ImageNet | CUB | Oxford Pets | DTD | Food101 | Place365 | Avg. |
|---|---|---|---|---|---|---|---|---|
| B/32 | WCA | 66.49 | 56.74 | 89.05 | 49.89 | 86.11 | 40.55 | 64.81 |
|  | BiFTA (w/o VF) | 66.51 | 58.11 | 88.56 | 51.27 | 86.41 | 41.80 | 65.44 |
|  | BiFTA (w/o DF) | 66.77 | 56.94 | 89.17 | 50.51 | 87.46 | 40.85 | 65.28 |
|  | BiFTA (ours) | 66.58 | 58.24 | 89.74 | 53.22 | 86.43 | 41.55 | **65.96** |
| B/16 | WCA | 71.05 | 59.87 | 92.13 | 52.87 | 89.99 | 41.33 | 67.87 |
|  | BiFTA (w/o VF) | 70.67 | 59.36 | 90.58 | 51.36 | 90.20 | 42.23 | 67.40 |
|  | BiFTA (w/o DF) | 71.10 | 59.91 | 91.83 | 53.56 | 90.38 | 42.11 | 68.15 |
|  | BiFTA (ours) | 71.14 | 60.06 | 91.67 | 54.64 | 90.11 | 42.12 | **68.29** |
| L/14 | WCA | 77.32 | 65.12 | 94.67 | 61.74 | 93.93 | 42.19 | 72.50 |
|  | BiFTA (w/o VF) | 77.14 | 65.46 | 94.63 | 62.09 | 93.94 | 42.51 | 72.63 |
|  | BiFTA (w/o DF) | 77.89 | 65.56 | 94.78 | 62.17 | 94.04 | 42.58 | 72.84 |
|  | BiFTA (ours) | 77.82 | 65.67 | 94.96 | 62.45 | 93.97 | 42.98 | **72.98** |

## F  Appendix 6: Limitation

One potential limitation we observed is that the textual descriptions generated by the LLM do not consistently focus on local features of the object. These descriptions often tend to be generic, making it difficult to associate specific parts of the text with corresponding local image patches from a human perspective. To address this, we aim to explore more advanced approaches to generate precise and localized descriptive texts that better align with the visual details of an image.

