# OpenReview forum: "\emph{Let's Roll a BiFTA}: Bi-refinement for Fine-grained Text-visual Alignment in Vision-Language Models"
_TMLR — Rejected by TMLR_

### Review · Reviewer_JaCU · 2025-04-01

**Summary Of Contributions:**

The paper introduces the BiFTA method, to improve the performance of CLIP in zero-shot scenarios. It is based on the WCA method, a method that extracts multiple random crops from an input image, classifies crops independently and aggregates the predictions to give an image-level output. The authors extend this method with additional textual supervision (inspired by CuPL and AttrVR), and two filtering mechanisms (one visual, one textual), to penalize extraction of overlapping visual crops and overlapping textual descriptions. The authors tested their method on six image classification benchmarks, reporting their results.

**Audience:**

Yes

**Broader Impact Concerns:**

No ethical concerns to report.

**Claims And Evidence:**

Yes

**Requested Changes:**

- The authors should expand on their method to differentiate themselves from the WCA method, exploring more the components they introduced, e.g., testing alternative strategies to select and filter the crops and to filter the textual descriptions, improving the gain from the additional textual guidance, etc.
- Add missing information regarding the crops extracted; they should also state explicitly that the method's hyperparameters are identical for all the datasets.
- Better explain the interplay between high IoU thresholds and a high number of extracted patches.

**Strengths And Weaknesses:**

Strength
- The presentation is clean and curated
- Marginal but consistent improvement in performance

Weaknesses
- The approach is extremely incremental, starting from an existing approach, and expanding it with additional textual guidance and two filtering mechanisms.
- The performance gain from these additions is limited, often with improvements lower than 1%.
- All experiments are based on the CLIP backbone (a pre-trained model from 2021) and do not consider more recent backbones, e.g., SigLIP (from 2023).
- While the proposed visual filtering strategy incentivizes diversity, it may not guarantee coverage of important visual parts (as visualized in Fig. 3). A selection strategy that incentivizes selecting subparts of the foreground object could have maybe provided better scores.
- The mathematical explanation of the posterior probability seems unnecessary.
- It is counterintuitive that having a high threshold value and a large number of extracted patches work better. By increasing both, shouldn’t there be a more strict limit on the number of non-overlapping patches one can extract? Moreover, the authors use a threshold of 0.8 for the IoU but do not report the total number of patches extracted, simply showing in Fig. 4 that the more the better.
- The authors state that their added textual guidance is suboptimal and unnecessary for the performance of their method. This means that the only technical improvement they propose is to filter the patches by reducing overlap.
- I am unsure why Fig. 5 report results averaged on 3 CLIP backbones.
- Alternative strategies to select and filter the patches should be considered to validate the effectiveness of the method, e.g., using a fixed grid without overlap, a fixed grid with partial overlap, etc.

---

> ### Author Response · Authors · 2025-04-20
> **Reply to the Review by Reviewer JaCU**
>
> We sincerely appreciate the valuable feedback from the reviewers. We provide detailed responses to address each point below.
>
> **Weakness 1** Result of different VLMs.
>
> **Reply 1:**
> Please refer to **Reply 2** of the rebuttal comment for **Reviewer 6wa2**.
>
> **Weakness 2** - The approach is extremely incremental ... often with improvements lower than 1%.
>
> **Reply 2:**
> First, we found the WCA method unintentionally generates redundant images and texts. Since the principle is to compute a cross-alignment score, the redundant views and descriptions can cause significant bias, which is discussed in Sec.4. BiFTA focuses on data refinement, where it can be incorporated with any cross-alignment methods. BiFTA demonstrates consistent and superior performance across multiple datasets without introducing computational overhead during inference.
>
> For zero-shot classification of CLIP, the presented improvement is already obvious[1][2], such as we achieved +1.50% improvement on CUB and +3.33% on DTD. The improvements remain compelling even on challenging task such as zero-shot classification on ImageNet-1K, we consistently outperform the baseline methods across all model sizes.
>
> [1] What does a platypus look like? Generating customized prompts for zero-shot image classification, ICCV 2023.
> [2] The Neglected Tails in Vision-Language Models, CVPR 2024.
>
> **Weakness 3** - While the proposed visual filtering strategy incentivizes diversity...
>
> **Reply 3:**
> In Fig. 3, we present the visualization result of the views and compare with the WCA. It shows that our View Refinement effectively eliminates redundant image patches while preserving most subparts of the foreground object. We understand that it might not cover all important visual parts due to the illustration only takes a small $N$ (Number of crop samples). The impact on the classification performance for $N$ in Section 5.3 further supports this phenomenon. The conclusion is the same as the reviewer's concern. When fewer subparts of the foreground object are captured, the zero-shot classification performance is worse.
>
> **Weakness 4** - The mathematical explanation of the posterior probability seems unnecessary.
>
> **Reply 4:**
> Thank you for your concerns. This theoretical section aims to demonstrate that BiFTA increases the number of effective views and descriptions. It provides a mathematical foundation for our intuitive approach. However, if you believe this section is unnecessary, we are open to moving it to the appendix. We believe this analysis is complementary to our empirical results, which helps to explain why the empirically observed gains occur. That said, we value clarity and focus; if the reviewer feels this section detracts from the main message, we are open to streamlining its presentation in the main text and moving the derivations to the appendix.
>
> **Weakness 5** - It is counterintuitive that having a high threshold value and a large number of extracted patches work better...
>
> **Reply 5:**
> For *"By increasing both, shouldn’t there be a more strict limit on the number of non-overlapping patches one can extract"*, this is indeed the fact. Please refer to the first paragraph of Sec. 5.3, where we explained in details. We understand that *"It is counterintuitive that having a high threshold value and a large number of extracted patches work better"*. In specific, a lower $\eta$ often results in a critic threshold where insufficient samples are unable to build up the patch queue. To address this, we re-sample from the queue until the required length $L$ is met, this could potentially lead to more redundant image patches in the patch queue. *"The authors use a threshold of 0.8 for the IoU but do not report the total number of patches extracted"*, we are sorry for the implicity. The number of patch settings are included in Sec. 5.1, where $n=60$. We will ensure it is further mentioned in Sec. 5.3 again.
>
> **Weakness 6** - The authors state that their added textual guidance is suboptimal and unnecessary...
>
> **Reply 6:**
> We are sorry about the implicit statement in the paper. The textual refinement actually leads to better performance. Appendix 5, Figure 5 and Table 10 clearly presents and compares the improvement with single modality refinement. Only the *RAG-based* refinement is suboptimal, which is clearly shown in Figure 5 and addressed in Appendix 4.
>
> **Weakness 7** - Alternative strategies to select and filter the patches should be considered ...
>
> **Reply 7:**
> Thanks for your valuable suggestion. We follow a consistent cropping method with WCA. Due to the time constraint, we would like you to view the result of some new refinement methods in Table 2 from **Reply 4, Reviewer 6wa2**. We will add evaluations of the methods you have mentioned in revised version.

---

### Review · Reviewer_6wa2 · 2025-04-05

**Summary Of Contributions:**

This paper introduces BiFTA, a method aims at addressing redundancy in both localized image patches and generated textual descriptions within WCA (ICML, 2024) for fine-grained visual-text alignment in vision-language models. BiFTA proposes two key refinements: view refinement, which eliminates overlapping image patches based on Intersection over Union (IoU) thresholds, and description refinement, which reduces redundant textual descriptions using cosine similarity based on the pretrained knowledge of CLIP. Experiments show that BiFTA improves zero-shot classification accuracy across six benchmark datasets, outperforming SoTA methods.

**Audience:**

Yes

**Broader Impact Concerns:**

None.

**Claims And Evidence:**

Yes

**Requested Changes:**

Major:

1.	Clarify novelty and technological contribution. Please emphasize the novelty and technological contributions of BiFTA, particularly in differentiating it from WCA.

2.	Extend experiments to other widely used VLMs. It is recommended to explore how different hyperparameters affect performance.

3.	Investigate hallucination issues. Investigate the influence of hallucination in LLM-generated descriptions.

4.	More ablation study. Please refer to the W4.

Minors:

1.	Correct the oversight of the 1/NM term in formula (1).

2.	It is recommended to replace model-generated images with natural images in figures, particularly for the title figure and model structure.

**Strengths And Weaknesses:**

Strengths

1.	This paper is well structured.

2.	The idea of this paper is intuitive and easy to understand.

3.	The experiments show mitigate such cropped images and text redundancy in WCA can enhancing the alignment of CLIP.

Weaknesses

1.	Limited technology contribution. The proposed BiFTA appears to be an incremental improvement over WCA [1]. The entire framework (i.e., cropped images and LLM-generated captions) is heavily based on WCA, with only minor adjustments, such as reducing redundant samples (e.g., repeated cropped images and LLM-generated captions). Therefore, I am concerning the technological contribution of this paper may not offer significant novelty to the community.

2.	Concerns about generalizability. The experiments are only conducted on a single pre-trained Vision-Language Model (VLM) (i.e., CLIP), with variations across backbones. Given that the hyperparameters of BiFTA (i.e., δ and ϵ) are crucial for performance (Refer the evidence in Figure 4), there are concerns about its generalizability to other widely-used open-source VLMs (e.g., BLIP [2,3] or ALIGN [4]). These parameters may require adjustment depending on the architecture and training set of different VLMs.

3.	Concerns about the potential hallucination issues. LLM’s description might contain information that is irrelevant to the content of the image. The description refinement stage forces the context of each description to be as inconsistent as possible. Therefore, I am concern that it might exacerbate the hallucination issues of LLM.

4.	Insufficient ablation study on view/description refinement. The ablation study does not explore alternative strategies for selecting redundant cropped images or descriptions. For example, what is the performance of leveraging CLIP’s knowledge for select redundant cropped images rather than IoU? What is the performance of leveraging simple strategies (e.g., TF-IDF) to select the repeated description?


References

[1] Li J, Li H, Erfani S, et al. Visual-text cross alignment: Refining the similarity score in vision-language models[J]. arXiv preprint arXiv:2406.02915, 2024.

[2] Li J, Li D, Xiong C, et al. Blip: Bootstrapping language-image pre-training for unified vision-language understanding and generation[C]//International conference on machine learning. PMLR, 2022: 12888-12900.

[3] Li J, Li D, Savarese S, et al. Blip-2: Bootstrapping language-image pre-training with frozen image encoders and large language models[C]//International conference on machine learning. PMLR, 2023: 19730-19742.

[4] Jia C, Yang Y, Xia Y, et al. Scaling up visual and vision-language representation learning with noisy text supervision[C]//International conference on machine learning. PMLR, 2021: 4904-4916.

---

> ### Author Response · Authors · 2025-04-20
> **Reply to the Review by Reviewer 6wa2**
>
> We sincerely appreciate the valuable feedback from the reviewer. We provide detailed responses to address each point below.
>
> **Change 1** Clarify novelty, contribution and difference from WCA.
>
> **Reply 1:**
> Contribution:
> First, we found the WCA method unintentionally generates redundant images and texts. Since the principle is to compute a cross-alignment score, the redundant views and descriptions can cause significant bias, which is discussed in Sec.4.
>
> We claim that redundant image patches are not always contributing to the correct classification, which is illustrated in Figure 1. Since the background features of the selected image are not meaningful in this case, the CLIP model may overemphasize the duplicated features so that skewing the result toward the redundant background image patches. Hence, BiFTA removes redundant views regarding the similarity metric and keeps more distinctive features.
>
> Also, We show that the original text descriptions contain redundant information in Figure 1. Ideally, we want to create a diverse set of text descriptions to local features of each image. This is hard since the descriptions generated by LLM are not comprehensive enough. In BiFTA, we incorporate a range of prompt templates from prior experiences that focus on describing both descriptive and distinctive attributes. Then, we leverage similarity metric to eliminate semantically similar texts and a top-k ranking so that it effectively eliminates redundant features that observed in existing methods (e.g., CuPL, WCA).
>
> Motivated by the discoveries, we investigated the impact of redundancy on cross-alignment methods. In Table 8, we clearly provided the results of a single view/text refinement can consistently outperform existing cross-alignment methods.
>
> Novelty:
> We sincerely acknowledge that BiFTA framework lacks certain novelty. However, we clearly claimed the existing redundancy issue and proposed BiFTA framework to improve CLIP zero-shot classification. We show a range of experiment results where **the results definitely support our claim**. We would like to remind you that the novelty or significance of methods is not the main selection [criteria of TMLR](https://shorturl.at/IoqDD).
>
> Difference from WCA:
> The BiFTA focuses on data refinement instead of proposing a similar classification methodology as WCA. Hence, our framework is a plug-and-play style module to incorporate with other visual-text involved cross-alignment methods for zero-shot classification.
>
> **Change 2** Extend experiments to other widely used VLMs.
>
> **Reply 2:**
> We select models such as ALIGN[1], AltCLIP[2], GroupViT[3] and SigLIP[4], the result is [here](https://shorturl.at/QXwdG). This consistency underscores the adaptability and effectiveness of our methodology when applied to a diverse range of VLMs.
>
>
> [1] Scaling up visual and vision-language representation learning with noisy text supervision, ICML 2021.
> [2] Altclip: Altering the language encoder in clip for
> extended language capabilities, ACL(Findings) 2022.
> [3] Groupvit: Semantic segmentation emerges from text supervision, CVPR 2022.
> [4] Sigmoid Loss for Language Image Pre-Training, ICCV 2023.
>
> **Change 3** - "LLM’s description might contain information that is irrelevant to the content of the image..."
>
> **Reply 3:**
> Since the prompt template is designed for each category rather than individuals, the generated descriptions on specific sample may vary in precision, which is addressed as hallucination problem. However, the inrelevant description $T_j$ will receive low text weight $v_j$ in WCA score function, thus effectively minimize their negative impact. Specifically, $v_j = \rm softmax(\rm cos(f_{txt}(\hat{T_y}),f_{txt}(T_j)))$, where $\hat{T_y}$ is the label prompt.
>
> Also, we draw inspiration from existing prompt templates and integrate them with our designed templates to enhance diversity. Unlike the fixed template used in WCA and CuPL, the new generated descriptions focusing on localized feature descriptions with new prompt templates. Also, we leverage a top-k ranking to select the most semantically relevant descriptions to further mitigate the hallucination issue, ensuring a robust corpus that strongly aligns with the target categories.
>
>
> **Change 4** - Additional ablation study.
>
> **Reply 4:**
> The result can be found [here](https://shorturl.at/Bj9ug)
>
> Notebly, the BiFTA(CLIP) shows slight improvement on the zero-shot classification, but it costs nearly 100x computational time than BiFTA(IoU) and scales exponentially with the increase of $N$ (Number of cropped samples).
> For Description Refinement, we notice that BiFTA(TF-IDF) outperforms BiFTA(CoS). The elimination of semantically similar text descriptions and top-k ranking are the keys to the performance so that we accept various ways of encoding.
>
> **Minor changes**
> 1. We are sorry about the inconsistency and will correct it in revised version.
> 2. We will add natural images in the title and model structure figures in revised version.

---

> ### Comment · Reviewer_6wa2 · 2025-05-05
>
> Thank you for the detailed rebuttal and additional results. While some concerns were addressed, my key concerns regarding limited technological contribution, hyperparameter sensitivity, and LLM hallucination remain unresolved.
>
> The main concern is the limited technology contribution, as BiFTA appears to be an incremental work of WCA [1], reusing its main components (cropped images and LLM-generated captions) with incremental filtering techniques (IoU and cosine similarity), which is also highlighted by other reviewers (i.e., JaCU and eXSx). This contribution, in my opinion, might not meet the bar for TMLR and may not be of enough interest to the TMLR audience.
>
> Moreover, although you show results across different backbones, all experiments remain within the WCA-style framework. BiFTA’s effectiveness beyond WCA is not demonstrated, making it unclear whether the observed gains are due to BiFTA or WCA itself, as evidenced by the marginal improvements in Table [https://anonymous.4open.science/r/TMLR-Supplementary-Tables-BF07/Table%201.pdf].
>
> I appreciate the effort and empirical rigor, but I believe addressing these points in a revised version of the paper would significantly enhance its impact.

---

### Review · Reviewer_qA1g · 2025-04-06

**Summary Of Contributions:**

This paper addresses the issue of redundant fine-grained text-vision alignment in CLIP-style models by proposing a semantic denoising method called BiFTA. The method consists of two components: view refinement and description refinement. For the image branch, it removes overlapping bounding boxes based on IoU scores. For the text branch, it filters out redundant descriptions using cosine similarity of embeddings and selects only the top-K descriptions most relevant to the class label.

The authors evaluate BiFTA on ViT-B/32 and ResNet-50/101, benchmarking its performance on downstream tasks including ImageNet, CUB, Oxford Pets, DTD, Food101, and Places365. Experimental results show that BiFTA consistently outperforms strong baselines such as WCA, CLIP-D, and CuPL.

**Audience:**

Yes

**Claims And Evidence:**

Yes

**Requested Changes:**

1. The paper lacks discussion and comparison with related methods that also aim to reduce redundancy in CLIP-style training, such as MaskCLIP and ACLIP.

[1] Maskclip: Masked self-distillation advances contrastive language-image pretraining, CVPR 2023.

[2] Attentive Mask CLIP, ICCV 2023.

**Strengths And Weaknesses:**

1. The paper tackles an important problem—how to enhance the performance of vision-language models by reducing redundancy in image-text alignment—which has clear practical value, especially in the context of modern VLMs.

2. The motivation is intuitive and well-grounded.

---

> ### Author Response · Authors · 2025-04-18
> **Reply to the Review by Reviewer qA1g**
>
> We sincerely appreciate the valuable feedback from the reviewer. We have carefully reviewed all comments and fully understand the concerns raised. Below, we provide detailed responses to address each point.
>
> **Changes 1** - The paper lacks discussion and comparison with related methods that also aim to reduce redundancy in CLIP-style training, such as MaskCLIP and A-CLIP.
>
> **Reply 1:**
> Thanks for your advice and we will ensure a complete discussion and comparison with the mentioned methods are included in the revised version.
>
> **MaskCLIP [1]** is a vision-language pre-training framework that incorporates a masked self-distillation technique into CLIP training stage, where our proposed BiFTA is an inference time data refinement framework. The major technical contribution of MaskCLIP is to facilitate the contrastive Vision-Language training process by leveraging a self-supervised learninng paradigm. The mask is to improve the transferability of the learned visual representations where the methods consistently join a shared and learnable feature vector across all inputs for the masked region and encoded the unmasked region individually. In addition, the principle between MaskCLIP and BiFTA is similar as we both hope the prediction can depend on the region containing distinctive features with less bias.
>
> **A-CLIP [2]** is an attentive masking technique which retaining a small number of image tokens that have a strong semantic correlation to the corresponding text description. The A-CLIP focuses on removing semantic irrelevant image tokens regarding the text description by introducing an attention-based score function. It contributes to efficient training of CLIP model during pre-training stage and achieves better zero-shot performance than original CLIP model and random masking method. On the other hand, our proposed BiFTA focuses on eliminating redundant image patches that randomly cropped from an entire image input while A-CLIP removes the image tokens when being processed within the vision encoder. Also, we both consider about the semantic relevance between the images and text description, where we compute a cross-alignment score. A-CLIP focuses on improving the efficiency and zero-shot performance during pre-training stage, while our BiFTA is a training-free framework that contributes to inference time improvement.
>
> [1] Maskclip: Masked self-distillation advances contrastive language-image pretraining, CVPR 2023.
>
> [2] Attentive Mask CLIP, ICCV 2023.

---

### Review · Reviewer_eXSx · 2025-04-15

**Summary Of Contributions:**

The paper proposes Bi-refinement for Fine-grained Text-visual Alignment (BiFTA), an algorithm built upon recent vision-language models—particularly the Weighted Cross Alignment (WCA) framework. BiFTA heuristically filters redundant image patches and textual descriptions to enhance vision language alignment. It consists of: 1) view refinement, which leverages Intersection-over-Union (IoU) thresholds to remove overlapping visual regions, and 2) description refinement, which filters semantically redundant text based on cosine similarity. As a result, the authors claim that BiFTA achieves superior zero-shot performance across a variety of datasets and backbone models.

**Audience:**

Yes

**Claims And Evidence:**

Yes

**Requested Changes:**

1. I find the following two statements somewhat confusing:
-  "The performance of CuPL shown in the chart is equivalent to only adapt view refinement of BiFTA. This suggests that refinement is enough to boost the zero-shot image classification performance"

--> However, it is unclear where the results corresponding to "view refinement only" from BiFTA are shown in Figure 5. Could you clarify this?

-  "The models often exhibit better performance with BiFTA w/o description refinements, which indicates our merged description set might not be an optimal description set."

--> In both statements, the proposed method (description refinement) appears to have a limited impact. It is also unclear whether it provides complementary benefits when combined with view refinement.


2. Besides the marginal performance improvements and weak motivation (solving the problem of WCA), more thorough ablation studies are necessary. While the simplicity of the method is not a weakness itself, it is important to justify why the current strategy was chosen over other simple alternatives.

**Strengths And Weaknesses:**

**Strengths**
1. The paper is well-structured and easy to follow, even for novice readers.
2. The related work and figures are presented intuitively, enhancing overall readability and comprehension.

**Weaknesses**
1. The main weakness lies in the marginal performance gains, which are coupled with a somewhat weak motivation and a relatively straightforward method that appears easily replaceable by other variants—yet without sufficient justification.
 - Namely, While I believe the simplicity of the method should not always be a drawback itself, the performance improvements of the proposed method of this paper seem marginal in most cases across datasets and backbones.
- More concrete ablation studies are necessary. The simplicity of the proposed approach calls for more rigorous and systematic comparisons. For example, why is IoU used for visual filtering and cosine similarity for textual filtering? Do the performance gains from description refinement stem from increased diversity in textual descriptions, or from filtering redundant content?  (I believe these two components should be separately assessed.)
- The method mainly targets the WCA method, which leverages local views and region-level features. However, it is unclear whether BiFTA can generalize to or benefit other recent vision-language models that do not explicitly rely on such components.

2. The purpose and necessity of the theoretical (posterior) analysis are unclear. This section appears to offer limited insight and may be redundant

---

> ### Author Response · Authors · 2025-04-20
> **Reply to the Review by Reviewer eXSx**
>
> We sincerely appreciate the valuable feedback from the reviewer. We have carefully reviewed all comments and fully understand the concerns raised. Below, we provide detailed responses to address each point.
>
> **Reply to Change 1:**
> *"It is unclear where the results corresponding to "view refinement only" from BiFTA are shown in Figure 5. Could you clarify this?"* We are sorry for the implicit explainations and we will remove the contents. In fact, we present a clear and detail comparison of BiFTA with single modality refinement in Table 8, Appendix E, where the result of BiFTA with only View Refinement is shown.
>
> *"In both statements, the proposed method (description refinement) appears to have a limited impact. It is also unclear whether it provides complementary benefits when combined with view refinement."* Also, this is included in Table 8, Appendix E, where we show that single/double view refinements exhibit predominant performance.
>
> **Reply to Change 2:**
> We clarify our contribution as follows:
> Contribution:
> First, we found the WCA method unintentionally generates redundant images and texts. Since the principle is to compute a cross-alignment score, the redundant views and descriptions can cause significant bias, which is discussed in Sec.4.
>
> We claim that redundant image patches are not always contributing to the correct classification, which is illustrated in Figure 1. Since the background features of the selected image are not meaningful in this case, the CLIP model may overemphasize the duplicated features so that skewing the result toward the redundant background image patches. Hence, BiFTA removes redundant views regarding the similarity metric and keeps more distinctive features.
>
> Also, We show that the original text descriptions contain redundant information in Figure 1. Ideally, we want to create a diverse set of text descriptions to local features of each image. This is hard since the descriptions generated by LLM are not comprehensive enough. In BiFTA, we incorporate a range of prompt templates from prior experiences that focus on describing both descriptive and distinctive attributes. Then, we leverage similarity metric to eliminate semantically similar texts and a top-k ranking so that it effectively eliminates redundant features that observed in existing methods (e.g., CuPL, WCA).
>
> Motivated by the discoveries, we investigated the impact of redundancy on cross-alignment methods. In Table 8, we clearly provided the results of a single view/text refinement can consistently outperform existing cross-alignment methods.
>
> The result of additional ablation studies can be found [here](https://shorturl.at/Bj9ug)
>
> Notebly, the BiFTA(CLIP) shows slight improvement on the zero-shot classification, but it costs nearly 100x computational time than BiFTA(IoU) and scales exponentially with the increase of $N$ (Number of cropped samples).
>
> For Description Refinement, we notice that BiFTA(TF-IDF) outperforms BiFTA(CoS). The elimination of semantically similar text descriptions and top-k ranking are the keys to the performance so that we accept various ways of encoding.

---

### Decision · Action_Editor_pyuc · 2025-05-17

**Recommendation:** Reject

**Comment:**

Overall the topic of submission is interesting, the paper is tackling an important problem that can have an impact in the community. The proposed solution is simple and elegant (which is not a weakness) -- namely filtering out redundant data. The reason I'm leaning to reject is aligned with the comments of the other reviewers.

The intuitions behind the paper are aligned with the current understanding in the community. That is on its own not a problem. But the methodology used leads to limited improvement. In such a situation, for the paper to have an impact in the community, and to be of interest to the reader, the authors need to (a) either support this more widely (e.g. test on other WCA-style framework etc.) or (b) provide in depth analysis (improve the scholarly element of the paper). E.g. explore robustness of hyper-parameters. Do a deep dive into the issue of hallucinations, etc. The goal of the detailed analysis and ablation is not to discover something non-intuitive, but rather to be thorough and hence the work to act as solid study of these intuitions.

I think without this thorough exploration, given the minor improvements and limited experiments and lack of an element of surprise, the paper will have limited impact in the community and therefore I concur with the majority of the reviewers that it might not be ready for acceptance. But I think a more thorough version of this work, a more detailed analysis, some additional ablation will be sufficient to push the paper towards an accept, so I encourage the authors to do any of those things and consider re-submission.

**Audience:**

Yes, the topic of the work is of interest to TMLR audience.

**Claims And Evidence:**

The claims provided in the paper are supported to a certain extent by empirical validation. Though I agree with the reviewers that more ablation or experiments are needed to fully support the claim.

**Resubmission Of Major Revision:**

The authors may consider submitting a major revision at a later time.